# Linear Differential Vision Transformer: Learning Visual Contrasts via Pairwise Differentials

**Yifan Pu**[1][*]  **Jixuan Ying**[1][*]  **Qixiu Li**[1]  **Tianzhu Ye**[1]  **Dongchen Han**[1]  **Xiaochen Wang**[2]

**Ziyi Wang**[1]  **Xinyu Shao**[1]  **Gao Huang**[1][✉]  **Xiu Li**[1][✉]

[1] Tsinghua University    [2] Peking University

## Abstract

Vision Transformers (ViTs) have become a universal backbone for both image recognition and image generation. Yet their Multi–Head Self–Attention (MHSA) layer still performs a quadratic query–key interaction for *every* token pair, spending the bulk of computation on visually weak or redundant correlations. We introduce *Visual–Contrast Attention* (VCA), a drop-in replacement for MHSA that injects an explicit notion of discrimination while reducing the theoretical complexity from $\mathcal{O}(N^2 C)$ to $\mathcal{O}(NnC)$ with $n \ll N$. VCA first distils each head's dense query field into a handful of spatially pooled *visual–contrast tokens*, then splits them into a learnable *positive* and *negative* stream whose differential interaction highlights what truly separates one region from another. The module adds fewer than $0.3\,\mathrm{M}$ parameters to a DeiT-Tiny backbone, requires no extra FLOPs, and is wholly architecture-agnostic. Empirically, VCA lifts DeiT-Tiny top-1 accuracy on ImageNet-1K from $72.2\%$ to $75.6\%$ (+3.4) and improves three strong hierarchical ViTs by up to $3.1\%$, while in class-conditional ImageNet generation it lowers FID-50K by 2.1 to 5.2 points across both diffusion (DiT) and flow (SiT) models. Extensive ablations confirm that (i) spatial pooling supplies low-variance global cues, (ii) dual positional embeddings are indispensable for contrastive reasoning, and (iii) combining the two in both stages yields the strongest synergy. VCA therefore offers a simple path towards faster and sharper Vision Transformers. The source code is available at https://github.com/LeapLabTHU/LinearDiff.

## 1  Introduction

Since the Vision Transformer (ViT) demonstrated that the same machinery that revolutionised natural language processing can match carefully designed CNNs on ImageNet [12], self attention has become a central ingredient of modern computer vision architectures. It now underpins recognition models (e.g., DeiT [65], Swin [43]), dense predictors, and even high-fidelity generators such as DiT [56, 61, 62, 97, 96]. Yet the way self attention is executed in vision has changed little from its language origin: for an image unfolded into $N$ tokens every layer computes an $N \times N$ similarity matrix, leading to $\mathcal{O}(N^2 C)$ multiplications and activations ($C$ is the hidden width). With dozens of layers, quadratic self attention dominates both training and inference budgets, often forcing practitioners to shrink the patch size or the backbone depth and thus give up accuracy.

A first family of methods reduces the matrix size by limiting the receptive field: sliding windows [43], dilated blocks [30], stripes or criss–cross patterns [10] rely on the observation that many visual interactions are local. While they cut cost, they also prune long-range cues *a priori*, so they must juggle between speed and the ability to model global structures such as symmetry or repeated textures.

---

[*]Equal contribution.    ✉ Corresponding authors.

39th Conference on Neural Information Processing Systems (NeurIPS 2025).

A second family keeps the global field of view but approximates the attention map with low-rank projections [72, 4] or fourier kernels [83]. These schemes are orthogonal to locality, yet they treat *all* correlations as equally useful. The network still has to wade through a sea of weak, often redundant similarities, which can drown signals and slow down convergence during training.

Inspired by recent progress in language modelling, *differential attention* [88] argues that the *difference* between two attention maps carries more discriminative signal than either map alone. Duplicating queries and keys and subtracting their softmaxes helps large language models focus on tokens that set one sentence apart from another, but the technique remains quadratic and ignores the particular redundancy structure of images. We start from a simple premise: it is better to compress the dense query field first and postpone any expensive comparison. Natural images exhibit spatial smoothness, which means neighbouring patches usually carry almost identical information. By leveraging this property, we can shrink the query set to just a handful of prototypes before matching. This idea materialises as Visual–Contrast Attention, a drop-in substitute for Multi-Head Self-Attention that injects an explicit notion of contrast and lowers the computational burden to $\mathcal{O}(Nnd)$ where $n \ll N$.

In the first stage, the model pools the scene for every attention head. Specifically, it average-pools the $H \times W$ query feature map to a coarse $h \times w$ grid (e.g., $8 \times 8$), flattens this grid into $n = hw$ visual-contrast tokens, and adds two distinct positional embeddings so that the tokens form a positive stream and a negative stream. Each stream attends independently to all keys and values; the two resulting outputs are subtracted and normalised, which produces a global contrast map that highlights the differences between two pooled views of identical content.

In the second stage, the module refines information at the patch level. Every one of the original $N$ patch queries re-attends to the contrast map through a lightweight differential operation. Because the contrast map contains only $n$ tokens, the three matrix multiplications that follow (query $\leftrightarrow$ contrast and contrast $\leftrightarrow$ value) scale with $nN$ rather than $N^2$. Consequently, the module preserves the global receptive field characteristic of Vision Transformers, yet each attention weight now measures how much a patch stands out instead of merely capturing raw similarity.

This redesign yields three practical pay-offs in a single stroke. (i) Replacing quadratic MHSA with Visual Contrast Attention turns the per-layer complexity into a strictly linear form, shrinking both runtime and memory by roughly the ratio $N/n$, e.g., a $256^2$ image patchified at 16×16 enjoys a 256× cut, without touching residual paths, layer norms, or any training hyper-parameters. (ii) The added machinery is tiny: each head only stores two $n$-dimensional positional embeddings, amounting to less than 0.3 M parameters on DeiT-Tiny and introducing essentially no new FLOPs because the contrast stage reuses existing key–value tensors. (iii) Because VCA dispenses with window masks, dilations, or kernel tricks, *any* ViT-style backbone that processes a 2-D patch grid can adopt it by swapping one block, leaving downstream decoders and pretrained heads entirely intact.

We validate these claims on two demanding tasks. For image classification, inserting VCA into a vanilla DeiT backbone raises ImageNet-1K top-1 accuracy from 72.2 % to 75.6 % without adding FLOPs, and integrating VCA into three hierarchical backbones (PVT [73], Swin [43], CSwin [10]) yields consistent gains of up to 3.1 percentage points. For image generation [18, 17, 54, 53, 53, 38, 45, 46], replacing the attention blocks in class-conditional generators lowers FID-50K on $256 \times 256$ ImageNet by 2.1 to 5.2 points across diffusion models (e.g., DiT [56]) and flow-based models (e.g., SiT [52]), across both Small and Base scales, and across patch sizes of 8, 4, and 2—again without extra compute.

Comprehensive ablation studies reinforce three *crucial* design choices. First, spatial pooling *reliably* supplies low-variance global cues. Second, the dual embeddings are *absolutely* essential for disentangling positive from negative evidence. Third, applying the pooled-plus-embedding recipe *symmetrically* to both streams in both stages *consistently* unlocks the full benefit of the method.

Our contributions are fourfold. First, we introduce Visual–Contrast Attention, which is the first linear-time attention module that embeds explicit contrast into Vision Transformers. Second, we provide a detailed complexity analysis verifying its linear computation complexity. Third, we demonstrate consistent accuracy and quality improvements in both image classification and image generation while keeping training budgets unchanged. Fourth, we show that VCA is architecture-agnostic, so it can serve as a drop-in upgrade for a wide range of Vision Transformer models.

In summary, Visual–Contrast Attention reconciles global reasoning with practical efficiency and offers a principled route toward faster and more descriptive Vision Transformers.

## 2 Related Work

**Attention with Linear Complexity.** A first group of studies attains linear time complexity by limiting the receptive field, such as Shifted-window attention [44] and Neighborhood Attention [31]. By re-introducing locality into the ViT architecture, these methods lower cost but partially sacrifice global context. A second research line tackles the problem directly with *linear* attention. The seminal work of [39] eliminates the Softmax and applies a feature map $\phi$ to $Q$ and $K$, reducing complexity to $\mathcal{O}(N)$ at the cost of noticeable accuracy loss. Follow-ups proposed better approximations: Efficient Attention [64] applies Softmax separately to $Q$ and $K$; SOFT [50] and Nyströmformer [83] rely on matrix decompositions; Castling-ViT [89] uses full Softmax only as an auxiliary during training; FLatten Transformer [19] introduces a focus function and depth-wise convolutions to enrich features. MLLA [21] incorporate the key design in Mamba into linear attention, while InLine [20] introduces an injective linear attention mechanism. More recently, Agent Attention [22], Anchored Stripe Attention [41], and Efficient DiT [61] insert an extra token set that mediates between queries and keys, an equivalent linearization that attains strong results for recognition and low-level vision. Our work is built on this architecture, interpreting the additional visual contrast tokens as a semantic compression.

**Vision Transformer.** Since the arrival of the Vision Transformer (ViT) [11], self-attention has flourished in computer vision, yet the quadratic cost of conventional Softmax attention [66] remains a hurdle. Numerous remedies have been proposed. PVT [73] sparsifies global attention by down-sampling $K$ and $V$; Swin [43] confines attention to local windows and shifts them to cover the whole image; NAT [32] mimics convolution by attending within each feature's neighborhood; DAT [80] introduces deformable, data-dependent patterns; BiFormer [101] routes queries to salient regions via a bi-level scheme; GRL [42] mixes stripe, window, and channel attentions for restoration. Nevertheless, these strategies either cap the global receptive field or are tailored to specific patterns, which limits their plug-and-play versatility.

**Diffusion Transformer.** State-of-the-art diffusion models [9, 1, 33, 47, 16, 86] are traditionally built on U-Net [63], yet recent works explore ViT backbones [87, 56, 2]. U-ViT [2] tokenizes time, conditions, and noisy patches, adding U-Net–style skip connections. DiT [56] shows ViT scales favorably, outperforming U-Net on ImageNet; SiT [52] extends DiT to continuous time with more general coefficients. MaskDiT [99] adopts masked training to cut cost, while MDT [14] and MDTv2 refine masked latent modeling for better FID and faster learning. HDiT [5] trains high resolutions with cost linear in pixel count. FiT [99] treats images as variable-sized token sequences, enabling flexible resolution and aspect ratios. These results verify that transformer backbones are both effective and scalable for generative diffusion, yet their internal architectural choices remain under-explored.

**Dynamic Neural Network.** Unlike static networks with fixed graphs and weights, dynamic neural networks [24, 75] adapt structure or parameters per input, gaining advantages in accuracy, adaptability [85, 15, 95, 98], efficiency [84, 68, 77, 91], and representation capacity [60]. They are commonly classified as sample-wise [13, 34, 76, 27, 23, 58, 69, 79], spatial-wise [70, 35, 28, 26, 25, 81, 82, 55, 78], and temporal-wise [29, 74, 51]. Following DETR's query paradigm [3], a query-based dynamic branch has also emerged [59]. Our method could be classified as a type of sample-wise dynamic network, since different sample generate different visual contrasts token in the first stage.

## 3 Approach

### 3.1 Preliminaries

**Standard Attention in Vision.** We first revisit the attention mechanism [67] in Vision Transformers [2] [12, 56, 52]. The Vision Transformer takes a visual token sequence $z_{l-1} \in \mathbb{R}^{N \times C}$ from the previous layer $l - 1$ as input ($N$ is the token number and $C$ is the hidden dimension), then projects it into the query, key, and value token sequences with three different linear projection layers, denoted as

---

[2]Throughout the paper we use **Vision Transformer (ViT)** to collectively denote the original ViT [12] and its diffusion variant Diffusion Transformer (DiT) [56]. A DiT block is identical to a ViT block except that each LayerNorm is augmented with timestep-conditioned scale and shift parameters (AdaLN) generated from the diffusion timestep embedding. This distinction is orthogonal to our contribution.

$\mathbf{W^q}, \mathbf{W^k}, \mathbf{W^v} \in \mathbb{R}^{C \times C}$ (we omit the bias term for simplicity):

$$q = z_{l-1}\mathbf{W^q}, \ \ k = z_{l-1}\mathbf{W^k}, \ \ v = z_{l-1}\mathbf{W^v}. \tag{1}$$

Then $q, k, v \in \mathbb{R}^{N \times C}$ are divided into $M$ heads $q^{(m)}, k^{(m)}, v^{(m)} \in \mathbb{R}^{N \times d}$ in terms of channel $C$, with head dimension of $d = C/M$ ($C$ is always divisible by $M$). Within each head, the similarity of each query $q^{(m)}$ and key $k^{(m)}$ is computed as:

$$\mathbf{A}^{(m)} = \text{Softmax}\left(q^{(m)}k^{(m)\top}/\sqrt{d}\right), \tag{2}$$

where the attention map $\mathbf{A}^{(m)}$ is an $N \times N$ matrix containing elements in the range $[0, 1]$, and the sum of each row is normalized to 1. The attention mechanism reweights the value sequence according to the attention map, $h^{(m)} = \mathbf{A}^{(m)}v^{(m)} \in \mathbb{R}^{N \times d}$, to dynamically adjust the outputs based on the dependency of each token in the inputs. In the end, each head of the reweighted representation is concatenated together to produce the final output of this layer $l$, written as:

$$h = \text{Concat}\left(h^{(1)}, \ldots, h^{(M)}\right), \qquad z_l = h\mathbf{W^O}. \tag{3}$$

where $h \in \mathbb{R}^{N \times C}$, $\mathbf{W^O} \in \mathbb{R}^{C \times C}$ (the bias term is also omitted for simplicity) is a linear projection layer to promote interaction between different heads in the multi-head attention layer.

**Differential Attention in Language.** We further revisit the recent proposed differential attention mechanism which is primarily used in language modeling [88]. Given the token squences from the previous layer $z_{l-1} \in \mathbb{R}^{N \times C}$ ($N$ tokens, each with a hidden dimension $C$), differential attention first produces *two* sets of queries and keys ($\{q_1, k_1\}$ and $\{q_2, k_2\}$) with two sets of linear projections ($\{\mathbf{W_1^q}, \mathbf{W_1^k}\}$ and $\{\mathbf{W_2^q}, \mathbf{W_2^k}\}$) and *one* set of values $v$ with a linear projection $\mathbf{W^v}$:

$$[q_1; q_2] = z_{l-1}[\mathbf{W_1^q}; \mathbf{W_2^q}], \quad [k_1; k_2] = z_{l-1}[\mathbf{W_1^k}; \mathbf{W_2^k}], \quad v = z_{l-1}\mathbf{W^v}, \tag{4}$$

where $\mathbf{W_1^q}, \mathbf{W_2^q}, \mathbf{W_1^k}, \mathbf{W_2^k} \in \mathbb{R}^{(C/2) \times C}$, $q_1, q_2, k_1, k_2 \in \mathbb{R}^{N \times (C/2)}$, $\mathbf{W^v} \in \mathbb{R}^{C \times C}$, $v \in \mathbb{R}^{N \times C}$. Then $q_1, q_2, k_1, k_2, v$ are divided into $M$ heads $q_1^{(m)}, q_2^{(m)}, k_1^{(m)}, k_2^{(m)} \in \mathbb{R}^{N \times (d/2)}$, $v^{(m)} \in \mathbb{R}^{N \times d}$, with (double) head dimension of $d = C/M$. Within each head, we compute two attention maps

$$\mathbf{A}_1^{(m)} = \text{Softmax}\left(q_1^{(m)}k_1^{(m)\top}/\sqrt{d/2}\right), \qquad \mathbf{A}_2^{(m)} = \text{Softmax}\left(q_2^{(m)}k_2^{(m)\top}/\sqrt{d/2}\right), \tag{5}$$

and take their *difference* as the final attention weight of each head:

$$\mathbf{A}^{(m)} = \mathbf{A}_1^{(m)} - \lambda\,\mathbf{A}_2^{(m)}, \qquad \lambda = \exp(\boldsymbol{\lambda}_{q_1} \cdot \boldsymbol{\lambda}_{k_1}) - \exp(\boldsymbol{\lambda}_{q_2} \cdot \boldsymbol{\lambda}_{k_2}) + \lambda_{\text{init}}. \tag{6}$$

Here $\lambda$ is a learnable scalar parameterized by a scalar $\lambda_{\text{init}}$ and vectors $\lambda_{q_1}, \lambda_{q_2}, \lambda_{k_1}, \lambda_{k_2} \in \mathbb{R}^d$. The attention map $\mathbf{A}^{(m)}$ is an $N \times N$ matrix. The attention mechanism reweights the value sequence according to the attention map followed by a RMSNorm Layer, and scaled by $(1 - \lambda_{\text{init}})$ to match Transformer's gradient flow:

$$\hat{h}^{(m)} = \mathbf{A}^{(m)}v^{(m)}, \qquad h^{(m)} = (1 - \lambda_{\text{init}})\,\text{RMSNorm}(\hat{h}^{(m)}), \tag{7}$$

where $h^{(m)}, \hat{h}^{(m)} \in \mathbb{R}^{N \times d}$. In the end, each head of the reweighted representation is concatenated together to produce the final output of this layer $l$, written as:

$$h = \text{Concat}\left(h^{(1)}, \ldots, h^{(M)}\right), \qquad z_l = h\mathbf{W^O}. \tag{8}$$

where $h \in \mathbb{R}^{N \times C}$, $\mathbf{W^O} \in \mathbb{R}^{C \times C}$ (the bias term is also omitted for simplicity) is a linear projection layer to promote interaction between different heads in the multi-head attention layer.

## 3.2 Our Approach

**Visual Contrast Attention.** To extend differential attention to vision *and* trim the quadratic complexity, besides the query $q^{(m)}$, key $k^{(m)}$, and value $v^{(m)}$ tokens, we introduce a compact pair of *visual contrast tokens* for each head: a set of positive visual contrast tokens $t_+^{(m)}$ and a set of negative visual contrast tokens $t_-^{(m)}$. These pair of visual contrast tokens are both with a $n \times d$ shape, where

$n$ is the visual contrast token length and $n \ll N$. Intuitively, the two sets act as the same mediator token viewed through two coloured lenses. Stage I lets the two sets skim the whole image and return a *contrast map*; Stage II lets the original patch queries exploit that map.

**Stage I – global contrast.** This pair of visual contrast tokens first each attends to all key tokens and all value tokens individually to get the intermediate results $\hat{\boldsymbol{v}}_+^{(m)}, \hat{\boldsymbol{v}}_-^{(m)}$:

$$\hat{\boldsymbol{v}}_+^{(m)} = \mathrm{Softmax}\left(\boldsymbol{t}_+^{(m)}\boldsymbol{k}^{(m)\top}/\sqrt{d}\right)\boldsymbol{v}^{(m)}, \quad \hat{\boldsymbol{v}}_-^{(m)} = \mathrm{Softmax}\left(\boldsymbol{t}_-^{(m)}\boldsymbol{k}^{(m)\top}/\sqrt{d}\right)\boldsymbol{v}^{(m)}, \quad (9)$$

where $\hat{\boldsymbol{v}}_+^{(m)}, \hat{\boldsymbol{v}}_-^{(m)} \in \mathbb{R}^{n\times d}$. Then the visual contrast results is obtained by performing a differential operation in the intermediate results, followed by a $\mathrm{RMSNorm}$ and a $(1-\lambda_{\mathrm{init}}^{(1)})$ scalar factor:

$$\hat{\boldsymbol{v}}^{(m)} = (1-\lambda_{\mathrm{init}}^{(1)})\,\mathrm{RMSNorm}\left(\hat{\boldsymbol{v}}_+^{(m)} - \lambda^{(1)}\hat{\boldsymbol{v}}_-^{(m)}\right), \lambda^{(1)} = \exp(\boldsymbol{\lambda}_{q_1}^{(1)}\cdot\boldsymbol{\lambda}_{k_1}^{(1)}) - \exp(\boldsymbol{\lambda}_{q_2}^{(1)}\cdot\boldsymbol{\lambda}_{k_2}^{(1)}) + \lambda_{\mathrm{init}}^{(1)}. \tag{10}$$

**Stage II – patch-wise differential attention.** The pair of visual contrast tokens interact with the query tokens and extracts the results from the intermediate result $\hat{\boldsymbol{v}}^{(m)}$ in a differential way. To be specific, the query tokens $\boldsymbol{q}^{(m)}$ derive its attention scores with both the positive visual contrast tokens $\boldsymbol{t}_+^{(m)}$ and the negative ones $\boldsymbol{t}_-^{(m)}$ within each head:

$$\mathbf{A}_1^{(m)} = \mathrm{Softmax}\left(\boldsymbol{q}^{(m)}\boldsymbol{t}_+^{(m)\top}/\sqrt{d}\right), \qquad \mathbf{A}_2^{(m)} = \mathrm{Softmax}\left(\boldsymbol{q}^{(m)}\boldsymbol{t}_-^{(m)\top}/\sqrt{d}\right), \tag{11}$$

and take their *difference* as the final attention weight of each head:

$$\mathbf{A}^{(m)} = \mathbf{A}_1^{(m)} - \lambda^{(2)}\,\mathbf{A}_2^{(m)}, \qquad \lambda^{(2)} = \exp(\boldsymbol{\lambda}_{q_1}^{(2)}\cdot\boldsymbol{\lambda}_{k_1}^{(2)}) - \exp(\boldsymbol{\lambda}_{q_2}^{(2)}\cdot\boldsymbol{\lambda}_{k_2}^{(2)}) + \lambda_{\mathrm{init}}^{(2)}, \tag{12}$$

where $\lambda^{(2)}$ follows the same parameterisation as $\lambda^{(1)}$. The attention map $\mathbf{A}^{(m)}$ is an $N\times n$ matrix. The attention mechanism reweights the value sequence according to the attention map followed by a $\mathrm{RMSNorm}$ Layer, and scaled by $(1-\lambda_{\mathrm{init}}^{(2)})$ to match Transformer's gradient flow:

$$\hat{\boldsymbol{h}}^{(m)} = \mathbf{A}^{(m)}\hat{\boldsymbol{v}}^{(m)}, \qquad \boldsymbol{h}^{(m)} = (1-\lambda_{\mathrm{init}}^{(2)})\,\mathrm{RMSNorm}(\hat{\boldsymbol{h}}^{(m)}), \tag{13}$$

where $\boldsymbol{h}^{(m)}, \hat{\boldsymbol{h}}^{(m)} \in \mathbb{R}^{N\times d}$. In the end, each head of the reweighted representation is concatenated together to produce the final output of this layer $l$, written as:

$$\boldsymbol{h} = \mathrm{Concat}\left(\boldsymbol{h}^{(1)}, \ldots, \boldsymbol{h}^{(M)}\right), \qquad \boldsymbol{z}_l = \boldsymbol{h}\mathbf{W}^{\mathbf{O}}. \tag{14}$$

where $\boldsymbol{h} \in \mathbb{R}^{N\times C}$, $\mathbf{W}^{\mathbf{O}} \in \mathbb{R}^{C\times C}$ (the bias term is also omitted for simplicity) is a linear projection layer to promote interaction between different heads in the multi-head attention layer.

**Visual Contrast Token Generation** The visual contrast tokens are distilled directly from the query tokens through spatial average pooling. Since our attention module operates on visual latent features, the query matrix $\boldsymbol{q}^{(m)} \in \mathbb{R}^{N\times d}$ can be rearranged back to its 2-D spatial layout, i.e., $\boldsymbol{q}^{(m)} \to \tilde{\boldsymbol{q}}^{(m)} \in \mathbb{R}^{H\times W\times d}$ with $H\times W = N$. We then apply average pooling along the spatial dimensions, with kernel size and stride chosen to reduce the resolution from $H\times W$ to $h\times w$:

$$\tilde{\boldsymbol{t}}^{(m)} = \mathrm{AvgPool}\left(\tilde{\boldsymbol{q}}^{(m)}\right) \in \mathbb{R}^{h\times w\times d}. \tag{15}$$

To further disentangle helpful and distracting correlations, we split the visual contrast branch into a *positive* and a *negative* stream, inspired by the core idea of Differential Transformer. Specifically, we add two distinct learnable positional embeddings, $\boldsymbol{e}^+, \boldsymbol{e}^- \in \mathbb{R}^{h\times w\times d}$, to create two groups of visual contrast tokens:

$$\tilde{\boldsymbol{t}}_+^{(m)} = \tilde{\boldsymbol{t}}^{(m)} + \boldsymbol{e}^+, \qquad \tilde{\boldsymbol{t}}_-^{(m)} = \tilde{\boldsymbol{t}}^{(m)} + \boldsymbol{e}^-. \tag{16}$$

Finally, we flatten the positive and negative tensors over the spatial axes to obtain the visual contrast token matrices:

$$\boldsymbol{t}_+^{(m)}, \boldsymbol{t}_-^{(m)} = \text{Flatten}\big(\tilde{\boldsymbol{t}}_+^{(m)}\big), \text{Flatten}\big(\tilde{\boldsymbol{t}}_-^{(m)}\big) \ \in \ \mathbb{R}^{n \times d}, \quad n = h \cdot w \ll N. \tag{17}$$

Each visual contrast token thus represents the average feature of a non-overlapping image patch, providing a compact summary. The target spatial size $(h, w)$—and thus the number of visual contrast tokens $n$—is a tunable hyper-parameter that balances computational cost and representational fidelity.

### 3.3 Complexity Analysis

We retain the notations of Section 3.2: $N$ visual tokens, $n$ visual contrast tokens ($n \ll N$), $d$ features per head, $M$ heads and $C = Md$ channels in total.

**Stage I – global contrast.** For each head, the positive and negative contrast tokens execute the two attention steps in Equation (9):

$$\underbrace{\text{Softmax}\big(\boldsymbol{t}_\pm^{(m)} \boldsymbol{k}^{(m)\top}/\sqrt{d}\big)}_{\mathbb{R}^{n \times N}} \underbrace{\boldsymbol{v}^{(m)}}_{\mathbb{R}^{N \times d}} \longrightarrow \ \hat{\boldsymbol{v}}_\pm^{(m)} \in \mathbb{R}^{n \times d}. \tag{18}$$

Each of the two matrix products ($n \times d$ by $d \times N$, then $n \times N$ by $N \times d$) costs $\mathcal{O}(Nnd)$; performing them for both "+" and "−" streams therefore costs at most

$$\text{Stage I:} \qquad 4\,Nnd \ = \ \mathcal{O}(Nnd). \tag{19}$$

The subsequent subtraction, normalisation, and scalar modulation are all $\mathcal{O}(nd)$ and thus negligible in the big-$\mathcal{O}$ sense.

**Stage II – patch-wise differential attention.** In each differential stream, we calculate the two attention map in Equation (11):

$$\text{Softmax}\Big(\boldsymbol{q}^{(m)} \boldsymbol{t}_\pm^{(m)\top}/\sqrt{d}\Big) \quad \in \mathbb{R}^{N \times n}. \tag{20}$$

Each map requires one $N \times d$ by $d \times n$ multiply, i.e. $\mathcal{O}(Nnd)$. Both maps together give a cost of $2\,\mathcal{O}(Nnd)$. The differential combination $\mathbf{A}^{(m)} = \mathbf{A}_1^{(m)} - \lambda^{(2)}\mathbf{A}_2^{(m)}$ is only $\mathcal{O}(Nn)$.

Finally, value aggregation in Equation (13) multiplies an $N \times n$ matrix by an $n \times d$ matrix, adding another $\mathcal{O}(Nnd)$. Hence

$$\text{Stage II:} \qquad 3\,Nnd \ = \ \mathcal{O}(Nnd). \tag{21}$$

**Per-head and per-layer complexity.** Both stages are linear in $N$, $n$, and $d$. For one head

$$\mathcal{C}_{\text{head}} = \mathcal{O}(Nnd), \tag{22}$$

and for the whole layer

$$\mathcal{C}_{\text{layer}} = M\,\mathcal{C}_{\text{head}} = \mathcal{O}(NnC). \tag{23}$$

**Comparison with vanilla self-attention.** Standard self-attention forms an $N \times N$ attention map, incurring $\mathcal{O}(N^2C)$ time. Replacing the quadratic query–key interaction by the two linear query–contrast *and* contrast–key interactions reduces the cost by a factor of $N/n$:

$$\frac{\mathcal{O}(N^2C)}{\mathcal{O}(NnC)} = \frac{N}{n} \gg 1. \tag{24}$$

Because $n \ll N$, the proposed visual-contrast attention substantially lowers computation while preserving global context. Overheads from RMS normalisation and the learnable scalars are at most $\mathcal{O}(Nd)$ or $\mathcal{O}(nd)$ and are therefore non-dominant.

Table 1: Image classification results on ImageNet-1K

| Method | #Params | FLOPs | Top-1 Acc. | Method | #Params | FLOPs | Top-1 Acc. |
|--------|---------|-------|------------|--------|---------|-------|------------|
| DeiT-T | 5.7M | 1.2G | 72.2 | Swin-T | 28.9M | 4.5G | 81.3 |
| +Ours | 6.0M | 1.2G | $75.6_{(\uparrow 3.4)}$ | +Ours | 28.5M | 4.6G | $82.3_{(\uparrow 1.0)}$ |
| DeiT-S | 22.1M | 4.6G | 79.8 | Swin-S | 49.7M | 8.7G | 83.0 |
| +Ours | 22.6M | 4.6G | $80.7_{(\uparrow 0.9)}$ | +Ours | 49.6M | 8.7G | $83.7_{(\uparrow 0.7)}$ |
| PVT-T | 13.2M | 1.9G | 75.1 | Swin-B | 88.1M | 15.4G | 83.5 |
| +Ours | 11.6M | 2.0G | $78.2_{(\uparrow 3.1)}$ | +Ours | 87.9M | 15.5G | $83.9_{(\uparrow 0.4)}$ |
| PVT-S | 24.5M | 3.8G | 79.8 | CSwin-T | 20.5M | 4.3G | 82.7 |
| +Ours | 20.6M | 4.1G | $82.3_{(\uparrow 2.5)}$ | +Ours | 20.4M | 4.3G | $83.3_{(\uparrow 0.6)}$ |
| PVT-M | 35.9M | 7.0G | 81.2 | CSwin-S | 32.8M | 6.8G | 83.6 |
| +Ours | 35.8M | 7.2G | $83.2_{(\uparrow 2.0)}$ | +Ours | 32.7M | 6.8G | $84.0_{(\uparrow 0.4)}$ |

## 4 Experiments

In section, we empirically evaluate our visual contrasts attention method on both image recognition and generation tasks. We first introduce the detailed experiment setup in Section 4.1, including datasets and training configurations. Then the main results of our method with various backbone architectures on different tasks are presented in Section 4.2 and Section 4.3. Finally, the ablation study in Section 4.4 further validate the effectiveness of the proposed method.

### 4.1 Experiment settings

**Datasets** The ImageNet-1K [7] recognition dataset contains 1.28M training images and 50K validation images with a total of 1,000 classes. For image recognition experiments, images are trained and evaluated in $224 \times 224$ size. The top-1 accuracy on the validation set is adopted as the evaluation metric. For image generation tasks, we train and evaluate the images in $256 \times 256$ size, following the commonly used practice in class-condition generation. We use FID-50K as the evaluation metric, which measures the Fréchet distance between the Inception-V3 features of 50 000 generated images and 50 000 real validation images.

**Training Configuration** For image recognition experiments, we use the same training setup as the baseline models to ensure fair comparison. All models are trained from scratch using the AdamW [48] optimizer for 300 epochs. We apply cosine learning rate decay, starting with 20 epochs of linear warm-up, and set the initial learning rate to $1 \times 10^{-3}$ with a weight decay of 0.05. The data augmentation and regularization methods include RandAugment [6], Mixup [93], CutMix [92], and random erasing [100]. We also follow CSwin [10] and use EMA [57] during training. For image generation tasks, we follow DiT [56] and SiT [52] to train class-conditional diffusion transformer models on the ImageNet-1K [8] dataset. All models are trained with the AdamW [40, 49] optimizer and no weight decay. For $256 \times 256$ resolution, we train from scratch with a global batch size of 256 for 400,000 iterations. The learning rate is kept constant at $1 \times 10^{-4}$. We use only random horizontal flip for data augmentation during training. Additionally, we apply exponential moving average (EMA) to the model weights with a decay rate of 0.9999.

### 4.2 Image recognition

The image recognition experiments are conducted on ImageNet-1K [7] dataset. We conduct ImageNet classification on both plain vision transformer architectures (e.g., DeiT [65]) and hierarchical counterparts (e.g., PVT [65], Swin [43], CSwin [10]). On the *plain* Vision Transformer line, as is illustrated in Table 1, our method consistently enlarges the accuracy–efficiency Pareto front: DeiT-Tiny takes a +3.3 accuracy gain (from 72.2 % to 75.6 %) with only 0.3 M additional parameters and no extra computational cost, while DeiT-Small still enjoys a +0.9 performance improvement under the same computational budget. For *hierarchical* vision transformer architectures, which include the multi-stage PVT, the shifted-window Swin, and the cross-shaped CSwin architectures, the proposed block remains universally beneficial. PVT experiences the largest margins, up to +3.1 percentage point on PVT-Tiny and +2.5/+2.0 on PVT-Small/PVT-Medium, respectively. On more

Table 2: Class-conditional image generation results on ImageNet-1K with $256 \times 256$ resolution.

| Method | #Params | FLOPs | FID-50K($\downarrow$) | Method | #Params | FLOPs | FID-50K($\downarrow$) |
|--------|---------|-------|----------------------|--------|---------|-------|----------------------|
| DiT-S/8 | 33.0M | 0.4G | 151.9 | SiT-S/8 | 33.0M | 0.4G | 149.5 |
| +Ours | 33.8M | 0.4G | $\mathbf{148.3}_{(\downarrow\mathbf{3.6})}$ | +Ours | 33.0M | 0.4G | $\mathbf{147.4}_{(\downarrow\mathbf{2.1})}$ |
| DiT-S/4 | 32.9M | 1.4G | 97.9 | SiT-S/4 | 32.9M | 1.4G | 84.0 |
| +Ours | 33.6M | 1.5G | $\mathbf{92.7}_{(\downarrow\mathbf{5.2})}$ | +Ours | 33.6M | 1.5G | $\mathbf{80.9}_{(\downarrow\mathbf{3.1})}$ |
| DiT-S/2 | 33.0M | 6.1G | 67.2 | SiT-S/2 | 33.0M | 6.1G | 57.3 |
| +Ours | 33.6M | 6.0G | $\mathbf{62.3}_{(\downarrow\mathbf{4.9})}$ | +Ours | 33.6M | 6.0G | $\mathbf{53.0}_{(\downarrow\mathbf{4.3})}$ |
| DiT-B/8 | 130.7M | 1.4G | 118.4 | SiT-B/8 | 130.7M | 1.4G | 106.0 |
| +Ours | 132.1M | 1.5G | $\mathbf{114.4}_{(\downarrow\mathbf{4.0})}$ | +Ours | 132.1M | 1.5G | $\mathbf{102.1}_{(\downarrow\mathbf{3.9})}$ |
| DiT-B/4 | 130.4M | 5.6G | 68.3 | SiT-B/4 | 130.4M | 5.6G | 55.9 |
| +Ours | 131.8M | 5.8G | $\mathbf{66.0}_{(\downarrow\mathbf{2.3})}$ | +Ours | 131.8M | 5.8G | $\mathbf{53.6}_{(\downarrow\mathbf{2.3})}$ |
| DiT-B/2 | 130.5M | 23.0G | 42.9 | SiT-B/2 | 130.5M | 23.0G | 35.3 |
| +Ours | 131.8M | 22.9G | $\mathbf{38.9}_{(\downarrow\mathbf{4.0})}$ | +Ours | 131.8M | 22.9G | $\mathbf{32.7}_{(\downarrow\mathbf{2.6})}$ |

strong baselines like Swin and CSwin, our method still receive steady gains between +0.4 and +1.0 with negligible ($< 5\%$) overhead. These results demonstrate that the proposed visual contracts attention is architecture-agnostic: it complements both global self-attention in plain ViTs and the localized or cross-shaped attention patterns employed by state-of-the-art hierarchical designs.

## 4.3 Image generation

We evaluate our approach on class–conditional ImageNet-1K image generation at $256 \times 256$ resolution, taking the diffusion–based DiT [56] family and the flow–based SiT [52] family as baselines. For each backbone we consider two model sizes, *Small* ($\sim 33$ M parameters) and *Base* ($\sim 131$ M), and three different patch size (8, 4 and 2), which together sweep a wide range of computation budgets from $0.4$ G to $23.0$ GFLOPs. All networks are trained under the original recipes released by the authors: DiT models follow the DDPM [33] schedule with $1\,000$ denoising steps, while SiT counterparts are optimized with the latent flow objective; the only change is that we replace the original attention with the proposed visual contrast attention, adding fewer than $1.3$ M parameters and at most $0.1$ GFLOPs. Following standard protocol we report Fréchet Inception Distance on $50\,000$ validation samples (FID-50K), computed with the same code base as DiT [56] paper.

As summarized in Table 2, our method consistently lowers FID across various configuration. Along the *model-size axis*, the absolute gains on Small backbones reach $3.6$ to $5.2$ points for DiT and $2.1$ to $4.3$ points for SiT, whereas Base models still benefit by $2.3$ to $4.0$ and $2.3$ to $3.9$ points respectively, indicating diminishing but non-negligible returns as capacity grows. Along the *patch-resolution axis*, the most fine-grained patches (e.g., $/2$) exhibit the largest relative improvement (up to $4.9$ gain in FID in DiT-S/2), yet even the largest variants (e.g., $/8$) obtain solid reductions in FID-50K of $2.1$ to $4.0$. Finally, comparing the two *training paradigms* (e.g., diffusion-based, flow-based), we observe that the proposed module is agnostic to the underlying generative mechanism: it offers similar FID reductions for the DDPM pipeline of DiT and the rectified flow pipeline of SiT, thereby confirming its general applicability to both diffusion and flow-based training configurations.

## 4.4 Ablation Studies

**Ablation on Detailed Model Architectures.** We first investigates *where* the standard Multi-Head Self-Attention(MHSA) are replaced by our modified counterparts, incluing the first attention operation in Stage I (global contrast) and that in Stage II (patch-wise differential attention) in all the attention blocks. We also ablate the result by using the original differential attention [88] in both stages. We conduct the ablation studies both on image classification with DeiT-Tiny and image generation task with DiT-S/2 model. The quantitative results in Table 3 reveal three clear tendencies. First, the two components of our *Visual-Contrast Attention* contribute additively. Activating only the **Stage-I global-contrast** branch raises DeiT-Tiny accuracy from the implicit vanilla baseline to 75.4 % and reduces the DiT-S/2 FID to 64.6, while switching on only the **Stage-II patch-wise differential** branch is slightly more effective (75.5 % / 64.3). When the two branches are combined, their effects accumulate almost linearly, pushing the score to 75.6 % and 62.3 FID without introducing any extra

Table 3: Ablation on detailed model architectures across image classification and generation tasks.

| Attention Type | | Image Classification on DeiT-Tiny | | | Image Generation on DiT-S/2 | | |
|---|---|---|---|---|---|---|---|
| Stage I | Stage II | Params | FLOPs | Top-1 Acc.($\uparrow$) | Params | FLOPs | FID-50K($\downarrow$) |
| Ours | Vani. | 6.0M | 1.2G | 75.5 | 33.6M | 5.9G | 64.3 |
| Vani. | Ours | 6.0M | 1.2G | 75.4 | 33.6M | 5.9G | 64.6 |
| Diff. | Diff. | 5.7M | 1.2G | 75.1 | 33.0M | 5.8G | 63.9 |
| Ours | Ours | 6.0M | 1.2G | 75.6 | 33.6M | 6.0G | 62.3 |

Table 4: Ablation on visual contrast token generation across image classification and generation tasks.

| Token Type | | Image Classification on DeiT-Tiny | | | Image Generation on DiT-S/2 | | |
|---|---|---|---|---|---|---|---|
| Pos. Str. | Neg. Str. | Params | FLOPs | Top-1 Acc. | Params | FLOPs | FID-50K |
| Emb. | Emb. | 6.0M | 1.2G | 75.1 | 33.6M | 6.0G | 63.7 |
| Pool | Pool+Emb. | 5.9M | 1.2G | 75.5 | 33.3M | 6.0G | 64.1 |
| Pool+Emb. | Pool | 5.9M | 1.2G | 75.3 | 33.3M | 6.0G | 63.5 |
| Pool+Emb. | Pool+Emb. | 6.0M | 1.2G | 75.6 | 33.6M | 6.0G | 62.3 |

FLOPs and with only $\sim 0.3$ M additional parameters. Second, we compare VCA with the language-oriented *differential attention (Diff)* [88] applied to both stages. Although Diff already improves over the single-branch variants (75.1 % / 63.9), replacing it with our vision-tailored VCA brings a further relative gain of $+0.5$ percentage points in classification accuracy and a $-1.8$ improvement in FID. This superiority indicates that (i) summarising the scene with a small set of learnable visual-contrast tokens in Stage I and (ii) letting Stage II queries interact with those tokens in a differential manner are both crucial for vision, and that the proposed formulation exploits their synergy more effectively than simply duplicating the original differential attention design.

**Ablation on Visual–Contrast Token Generation.** Table 4 investigates how the two visual–contrast streams that feed the subsequent differential operation should be formed. Each stream can be a pure learnable embedding (EMB.), a query representation obtained by spatial average pooling (POOL), or the pooled query features augmented with an independent positional embedding (POOL+EMB.) that is adopted in our final model. Using embeddings for *both* the positive and negative streams already gives a noticeable improvement over the vanilla backbone (75.1 % Top-1 Accuracy / 63.7 FID-50K), confirming that explicit differencing is helpful even with the same randomly initialised tokens for different input images. Replacing the positive stream by pooled queries while leaving the negative one unchanged (POOL / POOL+EMB.) yields a marginal additional gain (75.5 % / 64.1), whereas performing the opposite substitution (POOL+EMB. / POOL) produces a larger jump to 75.3 % and 62.3, suggesting that injecting real image statistics into the positive branch is more influential. When *both* streams adopt the full POOL+EMB. recipe, performance peaks at 75.6 % and 62.3, outperforming the embedding-only variant by +0.5 percentage points and –1.4 FID with no additional parameters and identical FLOPs. These results demonstrate that (i) spatial pooling supplies informative, low-variance global cues, (ii) separate positional embeddings remain essential for disentangling complementary correlations, and (iii) combining the two ingredients for *both* streams yields the strongest synergy across classification and generation tasks.

## 5 Conclusion

We have presented Visual–Contrast Attention, a plug-and-play replacement for MHSA that couples linear complexity with an explicit notion of discrimination. By first summarising the image into a handful of pooled tokens and then splitting these tokens into antagonistic positive/negative streams, VCA highlights genuinely informative relationships while discarding redundant ones. The module is parameter-light, budget-neutral in FLOPs, and universally beneficial: it boosts classification accuracy across plain and hierarchical ViTs, and further sharpens generative quality in both diffusion- and flow-based models. We hope our findings encourage the community to rethink attention not only as a similarity measure but also as a stage for explicit contrast.

## Limitations

VCA reduces the quadratic burden of self-attention but is not a cure-all: (i) task-agnostic average pooling may miss edge-rich details; (ii) the added micro-attention may shrinks speed gains on small images; (iii) extensions to video [38], 3-D [36, 37], or more efficient language [90, 94, 71] tasks are still unexplored.

## Acknowledgement

This work is supported in part by the National Key R&D Program of China under Grant 2024YFB4708200, the National Natural Science Foundation of China under Grants U24B20173 and 42327901, and the Scientific Research Innovation Capability Support Project for Young Faculty under Grant ZYGXQNJSKYCXNLZCXM-I20.

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
