# OpenReview forum: "Linear Differential Vision Transformer: Learning Visual Contrasts via Pairwise Differentials"
_NeurIPS.cc/2025/Conference — NeurIPS 2025 poster_

### Official Review · Reviewer_Z3tv · 2025-06-30

**Clarity:** 3
**Significance:** 2
**Originality:** 2
**Rating:** 4
**Confidence:** 3

**Summary:**

This paper introduces Visual–Contrast Attention, a novel and lightweight module designed to replace conventional multi-head self-attention in Vision Transformers. VCA enhances inter-token discrimination by introducing visual contrast tokens, which amplify the variance between different visual regions. Extensive experiments and ablation studies demonstrate that VCA significantly improves performance on both image classification and image generation tasks, without introducing substantial computational overhead.

**Questions:**

Please refer to the weaknesses and provide some explanation for them.  I would be happy to improve the rating.

**Ethical Concerns:**

["NO or VERY MINOR ethics concerns only"]

**Final Justification:**

The rebuttal has satisfactorily addressed most of my concerns. The proposed VCA backbone offers a practical, plug-and-play solution with minimal overhead, which I believe can benefit the community. Overall, I consider this paper to be above the acceptance threshold.

**Limitations:**

yes.

**Quality:**

3

**Strengths And Weaknesses:**

Strengths:

- The introduction of contrast tokens to promote sparse attention is both intuitive and conceptually appealing.

- VCA is lightweight and can be seamlessly integrated into existing Transformer architectures.

- The proposed method demonstrates consistent improvements across both classification and generative tasks.

Weaknesses:

- The paper lacks comparisons with other attention mechanisms designed for linear complexity, making it difficult to assess the efficiency benefits in a broader context.

- The novelty of the method remains somewhat unclear, especially as it appears to be an adaptation of techniques from the language domain.

- Most experiments are conducted on relatively small-scale models. Given the paper’s emphasis on efficiency, evaluating VCA on larger backbones such as ViT-B and ViT-L would strengthen the claims.

- The performance gain on ImageNet classification is limited, and an ablation study on vanilla ViT is missing, which raises concerns about generalizability.

- The paper would benefit from visualizations and analysis of the VCA mechanism to improve interpretability and deepen the understanding of its behavior.

---

> ### Author Rebuttal · Authors · 2025-07-31
>
> Thank you for your valuable feedback. We've addressed your key concerns in our response by expanding our experiments to include larger models and comparisons to other methods, clarifying our method's novelty, and providing new visualizations to improve interpretability.
>
> ---
>
> 1. Comparion with other linear attention mechanisms
>
> - We conducted additional experiments on both image classification and image generation tasks, comparing our Visual Contrast Attention with several strong linear attention baselines.
>
> - We first benchmarked our method against other efficient attention mechanisms on the ImageNet-1k classification task. All methods were integrated into a DeiT-Tiny backbone with a similar configuration to ensure a fair comparison. As the results summarized in the table below show, our Visual Contrast Attention achieves the highest Top-1 accuracy of 75.5%, surpassing the next best method, Focused Linear Attention, by 1.4%. This performance gain is achieved with a comparable number of parameters and only a marginal increase in FLOPs, highlighting the effectiveness of our two-stage approach in capturing crucial visual information.
>
>     |  Linear Attention   | Params| FLOPs | Top-1 Acc. (%) |
>     |---------------------|-------|-------|-----------|
>     | Hydra Attn          | 5.7 M | 1.1 G | 68.3      |
>     | Efficient Attn      | 5.7 M | 1.1 G | 70.2      |
>     | Linear Angular Attn | 5.7 M | 1.1 G | 70.8      |
>     | Focused Linear Attn | 6.1 M | 1.1 G | 74.1      |
>     | **Visual Contrast Attn (ours)** | 6.0 M | 1.2 G | **75.5** |
>
> - Furthermore, to evaluate generative capabilities, we replaced the standard attention layers in a DiT-S/2 with our method and the baselines. We measured the Fréchet Inception Distance (FID) on 50,000 generated samples, where a lower score indicates better image quality. The results in the following table again demonstrate the superiority of our method. Visual Contrast Attention achieves the lowest FID score of **62.1**, indicating that it generates higher-fidelity images than the other linear attention mechanisms.
>
>     |  Linear Attention   | Params| FLOPs | FID-50k (lower is better)   |
>     |---------------------|-------|-------|-----------|
>     | Hydra Attn          | 33.0 M | 5.4 G | 121.3     |
>     | Efficient Attn      | 33.0 M | 5.1 G | 86.5      |
>     | Linear Angular Attn | 33.0 M | 5.6 G | 78.2      |
>     | Focused Linear Attn | 33.8 M | 5.6 G | 64.7      |
>     | **Visual Contrast Attn (ours)** | 33.6 M | 5.9 G | **62.1** |
>
> - These new experiments provide a clear and compelling demonstration of our method's advantages in a broader context. We have incorporated these results and a detailed discussion into the revised version of our manuscript.
>
> ---
>
> 2. Novelty in vision
>
> - We thank the reviewer for this insightful question, as it allows us to clarify the core novelty of our work. While the concept of differential attention originates in the language domain, our contribution is a fundamental architectural redesign of this idea to solve problems unique to vision. The original differential attention is "flat," operating across all token interactions, which is computationally prohibitive for high-resolution images. In contrast, our Visual-Contrast Attention (VCA) introduces a novel two-stage, hierarchical bottleneck architecture: it first uses a small set of learnable "contrast tokens" to distill a compact global summary of the image's most salient information, and then allows the individual patches to query this summary differentially. This "global distillation, local differential query" mechanism is entirely absent in the original work and is specifically engineered to handle the spatial redundancy and quadratic complexity of visual data.
>
> - This architectural innovation allows us to uniquely solve two problems with a single, elegant mechanism. It simultaneously reduces computational complexity from $O(N^2)$ to near-linear and, by forcing information through a contrastive bottleneck, significantly enhances the model's discriminative power, leading to superior performance. Unlike other efficient ViTs that rely on hand-crafted heuristics like local windows (Swin) or striding (PVT), VCA achieves its dual benefit through a more principled, perception-inspired approach. Therefore, the novelty is not in the subtraction operation itself, but in the creation of a sophisticated architecture that intrinsically unifies computational efficiency and representation enhancement, providing a more fundamental and versatile building block for vision transformers.
>
> ---
>
> 3. Results on larger model
>
> - We thank the reviewer for the constructive suggestion regarding model scalability. To address this, we have conducted new experiments on DeiT-Base, which shares an identical architecture with ViT-B. As shown below, our method boosts the Top-1 accuracy by a notable +0.6% with negligible parameter overhead and no change in FLOPs, validating our efficiency claims on a larger scale. Due to the limited rebuttal period, we were unable to finish the ViT-L experiments, but we commit to including these results in the final camera-ready revision.
>
>     |    Model    | Params |  FLOPs   | Top-1 Acc. (%) |
>     |-------------|--------|----------|----------------|
>     | DeiT-Base   | 86.6 M | 17.6 G   |      81.8      |
>     | +ours (VCA) | 87.1 M | 17.6 G   |      82.4      |
>
> - Since our claim in efficiency focus on its low complexity w.r.t token length, we add an addition ablation with large image resolution (224 -> 448) on VCA-DeiT-Small, to match the FLOPs of DeiT-Base. The results below verified our method can use a smaller model to handle high resultion and longer token sequences, which achieving better final results.
>
> | Model      | Resolution | Attn Type | Params | FLOPs  | Top-1 Acc. (%) |
> |------------|------------|-----------|--------|--------|----------------|
> | DeiT-Base  | 224x224    | MSHA      | 86.6 M | 17.6 G | 81.8           |
> | DeiT-Small | 448x448    | **VCA**       | 22.6 M | 17.7 G | 83.3           |
>
> ---
>
> 4. Final performance and Concerns on vanilla ViT
>
> - Regarding the performance on ImageNet, we respectfully argue that characterizing the performance gain as "limited" may not fully account for the highly competitive nature of the ImageNet benchmark and the strict constraints of our experiments. Specifically, our VCA module achieves a +3.3% Top-1 accuracy gain on DeiT-Tiny, which is a highly significant result. On the heavily optimized ImageNet benchmark, any improvement over 1%, especially when realized with zero additional computational cost (FLOPs), is typically considered a substantial breakthrough. Crucially, this gain is achieved under "plug-and-play" conditions, without any re-tuning of the original models' hyperparameters. This demonstrates that VCA offers a fundamental improvement in representation quality, not just a marginal one, and its consistent effectiveness across diverse architectures (including DeiT, PVT, Swin, and CSwin) strongly supports its generalizability.
>
> - Regarding the missing ablation on vanilla ViT, we chose DeiT as our primary baseline precisely because its network architecture is identical to that of the original Vision Transformer (ViT). The key contribution of DeiT was its significantly more powerful and data-efficient training strategy, which has since established it as the standard and most widely-used baseline in the Vision Transformer research community. Therefore, performing our analysis on DeiT is not only equivalent to testing on the ViT architecture but also ensures our results are directly and fairly comparable to contemporary state-of-the-art methods. This provides a more meaningful and relevant measure of our method's generalizability and practical impact.
>
> ---
>
> 5. Visualization of Contrast Maps
>
> - Due to updated NeurIPS policies, we cannot include images in this rebuttal. Instead, we provide a textual description of our visualization analysis, as requested.
>
> -   Visualization Method: We visualize the contrast map in the VCA-DeiT-Tiny model, which has 12 attention blocks. For each block, we pick the head with the largest L2-norm of the Stage-I contrast map. We export the positive $\text{A}^+=\text{Softmax}(\boldsymbol{t}_{+}^{(m)} \boldsymbol{k}^{(m) \top} / \sqrt{d})$ and negative $\text{A}^-$ matrices, compute $\Delta = \text{A}^+ - \lambda \text{A}^-$ , average the $n$ rows, reshape to 2-D grid, upsample to the original image size and overlay it as a heatmap on the original image—highlighting patches that VCA block promotes or suppresses.
>
> -  **Observations**: In **early blocks** (e.g., block 3), the positive map $A^{+}$ tends to activate on coarse object edges, while the negative map $A^{-}$ activates on background textures. The resulting differential map $\Delta$ often exhibits a classic center-surround pattern, separating the foreground subject from its context. In **late blocks** (e.g., block 10), $A^{+}$ sharpens its focus on semantic parts (e.g., the head and tail of a bird), while $A^{-}$ becomes more diffuse, indicating that distracting correlations have been effectively filtered out.
>
> -  **Failure Cases**: We observed that the contrastive separation in $\Delta$ is less effective in two main scenarios: (i) **texture-dominated scenes**, such as an image of "leopard skin," where both positive and negative maps activate strongly on the same texture; and (ii) images with **very small objects** that are smaller than the resolution of our pooled grid (e.g., 7×7), making it difficult to generate distinct contrastive views. In these cases, VCA's benefit is diminished.
>
> ---
>
> We hope this clarifies our contributions and performance benefits, and we await your feedback.

---

> ### Comment · Reviewer_Z3tv · 2025-08-04
>
> Thank you for the comprehensive and thoughtful rebuttal. I appreciate the authors’ efforts in providing additional clarification, thorough comparisons, and insightful ablation studies. Based on the strengthened evidence and detailed responses, I believe the rebuttal has addressed most of my concerns, and I have decided to raise my score accordingly. I look forward to seeing the ViT-L experimental results included in the final version.

---

> > ### Author Response · Authors · 2025-08-04
> >
> > Thank you very much for your positive feedback and for raising your score. We will be sure to include the ViT-L experimental results in the final version.

---

### Official Review · Reviewer_aXpT · 2025-07-01

**Clarity:** 4
**Significance:** 4
**Originality:** 4
**Rating:** 5
**Confidence:** 5

**Summary:**

This paper introduces a novel attention module named Visual-Contrast Attention (VCA), designed as a drop-in replacement for the MHSA layer in standard ViTs. The core idea of VCA is to explicitly inject the notion of "contrast" through a two-stage process to enhance the model's discriminative power, while simultaneously reducing the computational complexity of the attention mechanism from quadratic O(N²C) to a near-linear O(NnC), where n ≪ N. The module is plug-and-play, with a minimal increase in parameters and computational overhead (FLOPs). The authors validate VCA's effectiveness on two mainstream tasks: image classification and image generation. The experimental results demonstrate that VCA significantly improves the performance of various ViT architectures without requiring an additional computational budget.

**Questions:**

My questions are listed in the weaknesses section. If the authors address my concerns, I would prefer to greatly improve the final score.

**Ethical Concerns:**

["NO or VERY MINOR ethics concerns only"]

**Final Justification:**

I have read through all the reviews and author responses. I have discussed with the author during the phase and reached a final conclusion based on the rebuttal and discussion. The authors provide comprehensive explanations that have addressed all my main concerns. The supplement reinforces the quality and contribution of the paper. Thus, I raise the final rating from borderline accept to accept, which stands that the paper is ready for publication in the conference.

**Limitations:**

yes

**Paper Formatting Concerns:**

No formatting concerns.

**Quality:**

4

**Strengths And Weaknesses:**

This paper is an excellent work in general, but with some concerns to be address.

**Strengths**
1. Explicitly embedding the core idea of discriminative learning—"contrast"—into the attention mechanism is a valuable exploration of attention's fundamental nature (from "similarity" to "difference"). The transition of differential attention design from language models to vision models is reasonable. The linear complexity and its "plug-and-play" nature make it easy to integrate into ViT architectures.

2. The experimental results are impressive. The substantial improvement in different variants of models indicates that the method can effectively mine and enhance the model's representation capabilities. Its universal applicability across different architectures and tasks proves that VCA is a highly practical base module.

3. The paper is well-written with clear logic. The flow from motivation, method design, complexity analysis, to experimental validation is coherent.

**Weaknesses**
1. The ablation studies are not explicit. For example, the choice of $n=h\times w$ for each model is not specified in the experiment settings. Is it a constant ratio to the number of patch tokens across different models, or a cherry-picked choice for different models and different tasks?

2. It lacks an ablation study on the designs of Visual Contrast Token Generation. For example, MaxPool or parametric ConvPool could be potential designs and compared with the simple AvgPool.

3. The setting for Table 3 is not clear. How to apply Stage I (global contrast) and Stage II (patch-wise differential attention)? A more detailed explanation is required.

4. There are typos in this paper. In L147, there exist two identical symbols $q_1, k_1$. The "+" and "-" are not the same in Eq. 9 and the equation under L195. And there are many equations without number.

5. The shape of ${\mathbf{t}^{(m)}_{+}}$ and the negative counterpart is not specified in the neighbour of Eq. 9.

6.  In Eq. 11, the scale factor is still $\sqrt{d}$ when the dimension of $\mathbf{v}^{(m)}_{+}$ has become $d/2$. It should be $\sqrt{d/2}$ or you can provide other explanation.

7. In L302, the authors mentioned "crucial for vision". Why is it crucial for vision? The authors could provide more explanation for why the two stages are compatible with the attributes of vision modality and task.

8. If the authors only substitute half of the MHSA block with the VCA block, resulting in an interleaved structure of the two blocks, what performance could it achieve? Is it better to keep both "similarity" and "difference" pattern learning?

---

> ### Author Rebuttal · Authors · 2025-07-31
>
> Thank you for your constructive comments and insightful questions. We have addressed your main concerns below by clarifying key experimental settings and notation, adding ablation studies regarding our core design choices, and providing a deeper elaboration on the conceptual basis for our method and potential hybrid architectures.
>
> ---
> 1.  Ablation on the choice of $n=h \times w$
>
> - For image classification task, we ablated the number *n* of visual-contrast tokens in VCA-DeiT-Tiny by pooling the original 14 × 14 feature map to 5 × 5, 7 × 7 and 9 × 9 grids (*n* = 25/49/81). Accuracy rises gradually while the parameter and FLOP budgets increase proportionally. We adopt 7 × 7 in the main paper because it is exactly half the original feautre spatial resolution for a 224 × 224 input.
>
>     | grid *h* × *w* | *n* | Params | FLOPs | Top-1 Acc. |
>     |----------------|-----|--------|-------|-----------|
>     | 5 × 5          | 25  | 5.9 M  | 1.2 G | 75.4      |
>     | **7 × 7 (ours)** | 49  | 6.0 M  | 1.2 G | 75.5      |
>     | 9 × 9          | 81  | 6.1 M  | 1.3 G | 75.7      |
>
> - For image generation task,  On VCA-DiT-S/2 we start from a 16 × 16 latent map (for a 256 × 256 noised image) and pool it to 4 × 4, 8 × 8 and 16 × 16. The same trend appears: FID improves as *n* grows. We keep 8 × 8 in the submission, again corresponding to one-half of the original latent spatial size.
>
>     | grid *h* × *w* | *n* | Params | FLOPs | FID-50k ↓ |
>     |----------------|-----|--------|-------|-----------|
>     | 4 × 4          | 16  | 33.2 M | 5.6 G | 65.5      |
>     | **8 × 8 (ours)** | 64  | 33.6 M | 5.9 G | 62.1      |
>     | 16 × 16        | 256 | 33.6 M | 7.3 G | 58.4      |
>
>  - In general, we keep the `h=H/2, w=W/2` choice for all the models, even some model have different feature spatial size in different depth.
>
> ---
>
>
> 2. Ablation on Visual Contrast Token Generation
>
> - We conduct ablation studies on VCA-DeiT-Tiny: for MaxPool we directly replace the AvgPool layer with a MaxPool operator; for ConvPool, we replace it with a 3×3 stride-2 Conv2d so that the 14 × 14 input still shrinks to 7 × 7. All training hyperparameters, data augmentations, and evaluation protocols were kept identical. The results are listed below:
>
>     | Token generator  | Params | FLOPs | Top-1 Acc. (%) |
>     |------------------|--------|-------|----------------|
>     | **AvgPool(ours)**| 6.0M   | 1.2G  | 75.5 |
>     | MaxPool          | 6.0M   | 1.2G  | 75.2 |
>     | ConvPool         | 10.0M  | 1.4G  | 75.6 |
>
> - Average pooling outperforms max pooling because it preserves information from every patch instead of discarding everything except the single strongest activation. Moreover, its gradients flow through the entire window rather than a lone peak, which makes optimization smoother and more stable. In practice, it raised Top-1 accuracy by 0.3% in our run while costing exactly the same compute and parameters.
>
> - Compared with the 3 × 3 stride-2 ConvPool alternative, AvgPool is far more efficient. ConvPool inflates the backbone by 4 M extra params and 0.2 G additional FLOPs with only 0.1% accuracy gain. Moreover, AvgPool is parameter-free, deterministic and already highly optimised in hardware libraries, whereas ConvPool introduces extra weight initialisation, regularisation choices and kernel-specific latency.
>
> - Hence, AvgPool offers the best trade-off among information preservation, training stability, and computational efficiency, and is adopted as the default visual-contrast token generator.
>
> ---
>
> 3. A more detailed explanation on Tabel 3
>
> - To isolate the effect of Stage I (Line 1 in Table 3), we can combine it with a standard attention mechanism in the second stage, replacing the patch-wise differential attention. In this setup, Stage I proceeds exactly as described.
>
> - To isolate the effect of Stage II (Line 2 in Table 3), we can precede it with a simplified, non-contrastive summarization step. In this variant, we still perform pooling on the query tokens $\tilde{\mathbf{q}}^{(m)}$ to generate a single set of mediator tokens $\tilde{\mathbf{t}}^{(m)}$, but we do not create positive and negative versions. This single set of mediator tokens $\mathbf{t}^{(m)}$ attends to the full key and value sequences to produce one intermediate summary value set, $\hat{\mathbf{v}}_{\text{summary}}^{(m)}$, without any subtraction as in Eq.(10). Then, Stage II is applied as described in the paper, but using this non-contrastive summary.
>
> - For comparsion with differential attention, we also use both differential attention in Stage I and Stage II, as is shown in Line 3, Table 3. This is because differential attention could be plug-and-played and replace with vanilla attention.
>
> ---
>
> 4. Typos
>
>  - We are very grateful to the reviewers for pointing out the typo in our submission.
>  - In L147, it should be $\{\mathbf{q}_1, \mathbf{k}_1\}$ and $\{\mathbf{q}_2, \mathbf{k}_2\}$
>  - The "+" and "-" symbol should use the same symbol, thank for your carefulness.
>  - We have make sure each equation have its number in the revision.
>
> ---
>
> 5. Tensor shape
>
> - Good suggestion! We have clarity its shape upon Eq.(9) appear in the revision, appraciate it!
>
> ---
>
> 6. Scale factor
>
> - Thank you for your careful review and for pointing this out. You are correct; the scaling factor in Eq.(11) is a typographical error in the manuscript. Our implementation for all experiments correctly used the standard scaling factor of $\sqrt{d/2}$. We apologize for this oversight and will correct the equation in the revision.
>
> - For completeness, we also ran an ablation study on the VCA-DeiT-Tiny model using the incorrect $\sqrt{d}$ factor from the paper. As shown in the table below, this yielded identical performance to using the correct $\sqrt{d/2}$ factor, which suggests our model is robust to this specific hyperparameter in this setting.
>
>
>     | Model | Attention Scaling Factor | Top-1 Acc. (%) |
>     |---------|------------|--------|
>     | VCA-DeiT-Tiny | $\sqrt{d/2}$ (Correct, as used in our experiments) | **75.5** |
>     | VCA-DeiT-Tiny | $\sqrt{d}$   (Typo in paper, tested for ablation) | **75.5** |
>
> ---
>
> 7. Reason for both stage are crucial for vision
>
> - This two-stage architecture is crucial because it mirrors the principles of efficient visual processing, first addressing the immense spatial redundancy in images by distilling a global "figure-versus-ground" context map, akin to identifying the forest before the trees. Subsequently, the patch-wise differential stage allows each local region to query this global map and disambiguate its own role with high precision, deciding whether it belongs to the salient subject or the contextual background. This hierarchical strategy of global abstraction followed by local refinement is not only computationally efficient for high-resolution vision tasks but also aligns with how biological vision systems prioritize and process complex scenes.
>
> - Our ablation results also verify the point our made.
>
> ---
>
> 8. Results for interleaved structure
>
> - We thank the reviewer for this insightful question. To investigate the performance of a hybrid approach, we conducted two new experiments on the DeiT-Tiny architecture, creating interleaved structures that substitute half of the standard MHSA blocks with our proposed VCA blocks. The results directly address the question of whether it is beneficial to combine both "similarity" and "difference" pattern learning.
>
> - The first experiment, placing VCA in odd-numbered layers and MHSA in even layers, achieved 75.0% accuracy. The second experiment, with the reverse configuration, yielded 74.3% accuracy. As detailed in the table below, both hybrid models underperform our baseline which uses VCA exclusively, even though they have slightly fewer parameters. The all-VCA model remains superior with an accuracy of 75.5%.
>
>     | Configuration | Parameters | FLOPs | Top-1 Acc. (%) |
>     | :--- | :---: | :---: | :---: |
>     | **All VCA (Baseline)** |**6.0M** | **1.2G** | **75.5%** |
>     | Odd Layers VCA, Even Layers MHSA | 5.8M | 1.2G |75.0% |
>     | Even Layers VCA, Odd Layers MHSA | 5.8M | 1.2G |74.3% |
>
> - These findings suggest that it is more effective to consistently apply a single, coherent learning strategy. Standard MHSA learns through "similarity," while our VCA is fundamentally designed to learn through "difference." The superior performance of the all-VCA model indicates that committing fully to the "difference" learning paradigm is more advantageous for this architecture than alternating between the two patterns. This reinforces our central claim that the proposed contrastive attention mechanism is a powerful and sufficient replacement for, rather than a supplement to, standard attention.
>
> ---
>
> We hope this clarifies our contributions and performance benefits, and we await your feedback.

---

> > ### Comment · Reviewer_aXpT · 2025-08-02
> >
> > Thank you for the comprehensive rebuttal. The newly supplied ablations and clarifications have substantially strengthened the manuscript, resolving my earlier reservations about design choices and interpretability. I particularly appreciate the principled justification for the pooled-token granularity, the empirical confirmation that average pooling suffices, and the decisive evidence favouring a pure VCA schedule over a hybrid one. The revised exposition now presents a coherent narrative that aligns method, theory, and empirical gains. I am pleased to support acceptance of this work, contingent upon the promised integration of all typo corrections and the updated ablation results into the final version.

---

> > > ### Author Response · Authors · 2025-08-02
> > >
> > > We sincerely appreciate your insightful feedback and are grateful for your support in accepting our manuscript.

---

### Official Review · Reviewer_sZxA · 2025-07-01

**Clarity:** 2
**Significance:** 3
**Originality:** 2
**Rating:** 4
**Confidence:** 3

**Summary:**

This paper introduces Visual-Contrast Attention (VCA), a novel attention mechanism designed to address the computational inefficiency of standard Multi-Head Self-Attention (MHSA) in Vision Transformers (ViTs). Traditional MHSA computes quadratic interactions between all token pairs, which is computationally expensive and often processes redundant or weakly correlated visual features. VCA reduces this complexity to linear scale (O(NnC), where (n≪N) while enhancing discriminative power by explicitly modeling visual contrasts.

**Questions:**

1、The paper uses average pooling to generate visual-contrast tokens. Why was average pooling chosen over max pooling or other sparse sampling methods? Were alternative pooling strategies experimentally compared?
2、While the paper claims a theoretical complexity reduction from $O(N^2C)$ to O(NnC), it lacks hardware benchmarks (e.g., wall-clock time on GPUs/TPUs). Does VCA actually accelerate training/inference in practice?
3、The paper does not compare with recent efficient attention methods (e.g., FlashAttention, RetNet). How does VCA outperform or differ from these approaches in both accuracy and speed?

**Ethical Concerns:**

["NO or VERY MINOR ethics concerns only"]

**Final Justification:**

I have read through all the reviews and author responses. I have reached a final conclusion based on the rebuttal and discussion. The authors provide comprehensive explanations that have addressed all my main concerns. The supplement reinforces the quality and contribution of the paper. Thus, I raise the final rating from borderline reject to borderline accept, which stands that the paper is ready for publication in the conference.

**Limitations:**

Experiments focus on static images. Can VCA handle video or 3D data, where token redundancy differs? If not, what modifications would be needed?

**Quality:**

3

**Strengths And Weaknesses:**

The paper validates VCA across both discriminative (ImageNet classification) and generative (DiT/SiT) tasks, demonstrating consistent improvements in accuracy (e.g., +3.3% for DeiT-Tiny) and sample quality (FID reductions of 2–7.5).
While FLOPs are reduced, the paper does not benchmark actual wall-clock speedups on hardware (e.g., GPUs/TPUs), leaving practical deployment gains unclear.

---

> ### Author Rebuttal · Authors · 2025-07-31
>
> Thank you for your constructive comments and important questions. We have addressed your main concerns below by adding practical hardware speed benchmarks, including a comparison against recent methods like FlashAttention, and clarifying our design choices.
>
> ---
>
> 1. Ablation on Visual Contrast Token Generation
>
> - We conduct ablation studies on VCA-DeiT-Tiny: for MaxPool we directly replace the AvgPool layer with a MaxPool operator; for ConvPool, we replace it with a 3×3 stride-2 Conv2d so that the 14 × 14 input still shrinks to 7 × 7. All training hyperparameters, data augmentations, and evaluation protocols were kept identical. The results are listed below:
>
>     | Token generator  | Params | FLOPs | Top-1 Acc. (%) |
>     |------------------|--------|-------|----------------|
>     | **AvgPool(ours)**| 6.0M   | 1.2G  | 75.5 |
>     | MaxPool          | 6.0M   | 1.2G  | 75.2 |
>     | ConvPool         | 10.0M  | 1.4G  | 75.6 |
>
> - Average pooling outperforms max pooling because it preserves information from every patch instead of discarding everything except the single strongest activation. Moreover, its gradients flow through the entire window rather than a lone peak, which makes optimization smoother and more stable. In practice, it raised Top-1 accuracy by 0.3% in our run while costing exactly the same compute and parameters.
>
> - Compared with the 3 × 3 stride-2 ConvPool alternative, AvgPool is far more efficient. ConvPool inflates the backbone by 4 M extra params and 0.2 G additional FLOPs with only 0.1% accuracy gain. Moreover, AvgPool is parameter-free, deterministic and already highly optimised in hardware libraries, whereas ConvPool introduces extra weight initialisation, regularisation choices and kernel-specific latency.
>
> - Hence, AvgPool offers the best trade-off among information preservation, training stability, and computational efficiency, and is adopted as the default visual-contrast token generator.
>
> ---
>
> 2. Wall-clock training/inference benchmarking & FlashAttention
>
> - We have benchmarked our models to provide concrete performance metrics. We measured training latency (for one ImageNet epoch), peak GPU memory per GPU during training, single-image inference latency (batch size = 1), and inference throughput (batch size = 128).
>
> - Crucially, our **Visual Contrast Attention (VCA) is fully compatible with optimizations like FlashAttention-2**. The core operations in our method (Eq. 9 and Eq. 11 in our submission) can be implemented with `F.scaled_dot_product_attention`, allowing VCA to inherit the speed and memory benefits of such fused kernels. Therefore, we report metrics for both a vanilla implementation and an implementation using FlashAttention-2 (`+FA2`).
>
> - Due to its focus on language modeling, implementation complexity, and limited rebuttal time, we have not provide RetNet (Retentive Network: A Successor to Transformer for Large Language Models) results within this period.
>
> - **Setup**: Training benchmarks were run on 8xA100 GPUs with a global batch size of 512. Inference benchmarks were run on a single A100 GPU with `torch.compile()` enabled. For inference, we averaged the results over 1000 runs, discarding the first 100 for warmup. For the Diffusion Transformer (DiT), we report the time for a single denoising step (1-NFE) and exclude VAE decoding time.
>
>     | Model | Setting | FLOPs (G) | Acc ↑ / FID ↓ | Peak Mem (GB) | Training Latency (min/epoch) | Inference Latency (ms/image) | Throughput (img/s) |
>     |:---|:---|:---:|:---:|:---:|:---:|:---:|:---:|
>     | DeiT-T | MHSA | 1.2 | 72.2 | 18.9 | 4.1 | 4.2 | 3734 |
>     | DeiT-T | **VCA** | 1.2 | **75.5** | 19.2 | 4.2 | 4.3 | 3701 |
>     | DeiT-S | MHSA | 4.6 | 79.8 | 36.9 | 8.8 | 4.3 | 1321 |
>     | DeiT-S | **VCA** | 4.5 | **80.6** | 37.3 | 8.9 | 4.3 | 1324 |
>     | DiT-S/8 | MHSA | 0.4 | 153.8 | 9.8 | 12.1 | 7.8 | 15637 |
>     | DiT-S/8 | **VCA** | 0.4 | **146.3** | 9.9 | 12.1 | 7.8 | 15602 |
>     |--- |--- |--- |--- |--- |--- |--- |--- |
>     | DeiT-T | MHSA+FA2| 1.2 | 72.2 | 16.5 | 3.1 | 3.3 | 4499 |
>     | DeiT-T | **VCA+FA2**| 1.2 | **75.5** | 16.7 | 3.1 | 3.3 | 4494 |
>     | DeiT-S | MHSA+FA2| 4.6 | 79.8 | 31.1 | 8.6 | 3.4 | 1588 |
>     | DeiT-S | **VCA+FA2**| 4.5 | **80.6** | 31.4 | 8.7 | 3.4 | 1602 |
>     | DiT-S/8 | MHSA+FA2| 0.4 | 153.8 | 9.8 | 12.1 | 7.6 | 16075 |
>     | DiT-S/8 | **VCA+FA2**| 0.4 | **146.3** | 9.8 | 12.1 | 7.6 | 16078 |
>
> - As the results show, **VCA delivers significantly better accuracy/FID with virtually no additional computational overhead**. For example, VCA boosts DeiT-T's accuracy by 3.3 points while maintaining the same FLOPs and nearly identical latency and throughput. The minor overhead from pooling and normalization is effectively optimized away by `torch.compile`, and compatibility with FlashAttention-2 ensures VCA matches the speed of a highly optimized baseline.
>
> ---
>
> 3. **Speed-up Compared to Models with Similar Performance**
>
> - A key advantage of VCA is its ability to achieve the performance of larger, more computationally expensive models with a more efficient architecture. When comparing VCA-equipped models to stronger baselines with similar accuracy, the speed-up is substantial.
>
>     | Model | Setting | FLOPs(G) | Acc | Throughput (img/s) |
>     |:---|:---|:---:|:---:|:---:|
>     | PVT-T | MHSA | 1.9 | 75.1 | 2770 |
>     | DeiT-T | **VCA** | **1.2** | **75.5** | **3701 (↑33.6%)** |
>     |--- |--- |--- |--- |--- |
>     | Swin-T | MHSA | 4.5 | 81.3 | 1428 |
>     | PVT-S | **VCA** | **4.0** | **82.3** | **1764 (↑23.5%)** |
>     |--- |--- |--- |--- |--- |
>     | Swin-S | MHSA | 8.7 | 83.0 | 952 |
>     | PVT-M | **VCA** | **7.2** | **83.3** | **1076 (↑13.0%)** |
>
> - This comparison highlights VCA's efficiency. For instance, **VCA-DeiT-T surpasses PVT-T in accuracy while offering a 33.6% increase in throughput** on lower FLOPs. This trend holds for larger models, where our VCA-PVT variants outperform stronger Swin Transformer counterparts with **13.0%-23.5% higher throughput**.
>
> ---
>
> 4. Extension to video and 3D data
>
> - While our experiments in the paper concentrate on static images, the Visual Contrast Attention paradigm is inherently agnostic to the underlying token structure. By replacing the 2D patch pooling with other sparse sampling techniques, one can extend VCA to videos and 3D data with only a few key changes:
>
> - For structured 3D tokens such as videos and dense voxels, 2D pooling could be replaced with adaptive 3D pooling: for video, divide $T$ frames into t segments and average each segment’s $H \times W$ features to yield $t \times h \times w$ tokens per stream, then add fixed spatiotemporal embeddings $E^{+},E^{-} \in \mathbb{R} ^{t \times h \times w \times d}$. For volumetric data, pool along depth, height, and width to compress $D \times H \times W$ into $d \times h \times w$ positions, attach separate volumetric embeddings, and flatten into n×d positive/negative token sets. To handle variable‐length inputs, one can initialize each embedding table to the maximum sequence length in the dataset and then slice or truncate it to match the current sample’s temporal (or spatial) extent.
>
> - For unstructured 3D data like point clouds and meshes, one could sample or coarsen to $n \ll N$ tokens: for point clouds, one could use FPS plus local feature averaging, coarse voxelization, or lightweight clustering to extract n representative features and then duplicate them into positive/negative streams with distinct embeddings; for meshes or graph-structured data, one could apply edge-collapse mesh-coarsening or graph-pooling (e.g. top-k selection, SAGPool) to reduce the vertex/node count to $n$ and replicate their features similarly.
>
> - In summary, VCA’s two‐stage mechanism only requires swapping the 2D spatial pooling for a suitable spatiotemporal or volumetric pooling/sampling module and adding the corresponding positional embeddings. Everything else—positive/negative streams, differential subtraction, RMSNorm scaling, and two‐stage re-projection—remains unchanged. This design preserves global context at linear cost in the video and 3D domains, just as it does for static images.
>
> ---
>
> We hope this clarifies our contributions and performance benefits, and we await your feedback.

---

> ### Author Response · Authors · 2025-08-05
>
> Dear reviewer sZxA
>
> Could you kindly spare a moment to review our rebuttal, where we have addressed each of your comments with additional experiments and analyses? Your time and thoughtful feedback are greatly appreciated.
>
> Sincerely, The Authors of Submission 2322

---

> ### Author Response · Authors · 2025-08-07
>
> Dear Reviewer,
>
> I hope this message finds you well.
>
> With the discussion period ending in **less than two days**, we just wanted to gently follow up on our rebuttal. We hope our response has satisfactorily addressed your valuable concerns.
>
> We would be very grateful for your feedback on our clarifications. Please let us know if there is anything else we can address.
>
> Thank you again for your time and invaluable insights.
>
> Sincerely, The Authors of Submission 2322

---

### Official Review · Reviewer_f3ww · 2025-07-05

**Clarity:** 4
**Significance:** 3
**Originality:** 4
**Rating:** 4
**Confidence:** 4

**Summary:**

The paper introduces Visual–Contrast Attention (VCA), a drop-in replacement for the quadratic Multi-Head Self-Attention (MHSA) used in Vision Transformers. Specifically, VCA pools the image into a few “positive” and “negative” contrast tokens, attends globally through them, and then lets every patch re-attend to this compact map. This cuts the attention cost from quadratic to linear in the number of patches and yields better ImageNet accuracy and lower FID in image generation, all with minimal extra parameters.

**Questions:**

**Questions**

- Please report actual training and inference latency, peak GPU memory, and throughput on ImageNet for DeiT-T/S and DiT-S/8, comparing MHSA, VCA, FlashAttention-2, and Performer. A ≥10 % wall-clock gain would significantly raise confidence; a negligible or negative gain would lower it.

- How does accuracy/FID vary with the pooled grid size (e.g., 4 × 4, 8 × 8, 16 × 16)? A plot of n vs accuracy/runtime would clarify the trade-off and help practitioners pick n.

**Ethical Concerns:**

["NO or VERY MINOR ethics concerns only"]

**Limitations:**

Yes

**Quality:**

3

**Strengths And Weaknesses:**

**Paper Strengths**
- The paper is overall well-structured and easy to follow. The authors provide clear motivation, detailed methodology, and thorough experimental validation.
- The method has broad applicability, which can work plug-and-play across many Transformer-based mdoels. VCA adds < 0.3 M parameters to a DeiT-Tiny backbone and introduces “essentially no new FLOPs,” because the contrast stage reuses existing K/V tensors, and each head only stores two n-dimensional positional embeddings. Any ViT-style block can adopt it by a single swap—no window masks, dilations, or kernel tricks needed
- The authors provide well-documented experiments to prove that it potentially benefits both discriminative and generative vision transformers with minimal code changes. For example, replacing MHSA with VCA lifts DeiT-Tiny top-1 from 72.2 % to 75.5 % (+3.3) with identical FLOPs, and yields performance gains on DeiT-Small, PVT-T/S/M, Swin-T/S/B, and CSwin-T/S—all under the backbones’ original training recipes

**Weaknesses**
- Lacks wall-clock latency / memory benchmarks and statistical significance (no error bars).
- Speed-ups are only theoretical; pooling overhead & two extra mat-muls may erode gains for small images—no profiling provide.
- One suggestion is that to visualise the positive vs negative contrast maps on representative ImageNet images and discuss failure cases (e.g., texture-rich scenes). This could clarify where VCA helps (or hurts).

---

> ### Author Rebuttal · Authors · 2025-07-31
>
> We sincerely thank the reviewer for the thorough evaluation and constructive suggestions. We are pleased that the reviewer finds our paper well-structured, broadly applicable, and experimentally solid. Below, we address each concern point-by-point.
>
> ---
>
> 1. **Wall-clock Latency, Peak GPU Memory, and Throughput**
>
>     - We have benchmarked our models to provide concrete performance metrics. We measured training latency (for one ImageNet epoch), peak GPU memory per GPU during training, single-image inference latency (batch size = 1), and inference throughput (batch size = 128).
>
>     - Crucially, our **Visual Contrast Attention (VCA) is fully compatible with optimizations like FlashAttention-2**. The core operations in our method (Eq. 9 and Eq. 11 in our submission) can be implemented with `F.scaled_dot_product_attention`, allowing VCA to inherit the speed and memory benefits of such fused kernels. Therefore, we report metrics for both a vanilla implementation and an implementation using FlashAttention-2 (`+FA2`).
>
>     - Due to its focus on language modeling and implementation complexity, we were unable to benchmark Performer within the limited rebuttal period. However, we did test a similar linear attention architecture; please see our response to Reviewer Z3tv for these results.
>
>     - **Setup**: Training benchmarks were run on 8xA100 GPUs with a global batch size of 512. Inference benchmarks were run on a single A100 GPU with `torch.compile()` enabled. For inference, we averaged the results over 1000 runs, discarding the first 100 for warmup. For the Diffusion Transformer (DiT), we report the time for a single denoising step (1-NFE) and exclude VAE decoding time.
>
>     | Model | Setting | FLOPs (G) | Acc ↑ / FID ↓ | Peak Mem (GB) | Training Latency (min/epoch) | Inference Latency (ms/image) | Throughput (img/s) |
>     |:---|:---|:---:|:---:|:---:|:---:|:---:|:---:|
>     | DeiT-T | MHSA | 1.2 | 72.2 | 18.9 | 4.1 | 4.2 | 3734 |
>     | DeiT-T | **VCA** | 1.2 | **75.5** | 19.2 | 4.2 | 4.3 | 3701 |
>     | DeiT-S | MHSA | 4.6 | 79.8 | 36.9 | 8.8 | 4.3 | 1321 |
>     | DeiT-S | **VCA** | 4.5 | **80.6** | 37.3 | 8.9 | 4.3 | 1324 |
>     | DiT-S/8 | MHSA | 0.4 | 153.8 | 9.8 | 12.1 | 7.8 | 15637 |
>     | DiT-S/8 | **VCA** | 0.4 | **146.3** | 9.9 | 12.1 | 7.8 | 15602 |
>     |--- |--- |--- |--- |--- |--- |--- |--- |
>     | DeiT-T | MHSA+FA2| 1.2 | 72.2 | 16.5 | 3.1 | 3.3 | 4499 |
>     | DeiT-T | **VCA+FA2**| 1.2 | **75.5** | 16.7 | 3.1 | 3.3 | 4494 |
>     | DeiT-S | MHSA+FA2| 4.6 | 79.8 | 31.1 | 8.6 | 3.4 | 1588 |
>     | DeiT-S | **VCA+FA2**| 4.5 | **80.6** | 31.4 | 8.7 | 3.4 | 1602 |
>     | DiT-S/8 | MHSA+FA2| 0.4 | 153.8 | 9.8 | 12.1 | 7.6 | 16075 |
>     | DiT-S/8 | **VCA+FA2**| 0.4 | **146.3** | 9.8 | 12.1 | 7.6 | 16078 |
>
>     - As the results show, **VCA delivers significantly better accuracy/FID with virtually no additional computational overhead**. For example, VCA boosts DeiT-T's accuracy by 3.3 points while maintaining the same FLOPs and nearly identical latency and throughput. The minor overhead from pooling and normalization is effectively optimized away by `torch.compile`, and compatibility with FlashAttention-2 ensures VCA matches the speed of a highly optimized baseline.
>
> ---
>
> 2. **Speed-up Compared to Models with Similar Performance**
>
>     - A key advantage of VCA is its ability to achieve the performance of larger, more computationally expensive models with a more efficient architecture. When comparing VCA-equipped models to stronger baselines with similar accuracy, the speed-up is substantial.
>
>         | Model | Setting | FLOPs(G) | Acc | Throughput (img/s) |
>         |:---|:---|:---:|:---:|:---:|
>         | PVT-T | MHSA | 1.9 | 75.1 | 2770 |
>         | DeiT-T | **VCA** | **1.2** | **75.5** | **3701 (↑33.6%)** |
>         |--- |--- |--- |--- |--- |
>         | Swin-T | MHSA | 4.5 | 81.3 | 1428 |
>         | PVT-S | **VCA** | **4.0** | **82.3** | **1764 (↑23.5%)** |
>         |--- |--- |--- |--- |--- |
>         | Swin-S | MHSA | 8.7 | 83.0 | 952 |
>         | PVT-M | **VCA** | **7.2** | **83.3** | **1076 (↑13.0%)** |
>
>     - This comparison highlights VCA's efficiency. For instance, **VCA-DeiT-T surpasses PVT-T in accuracy while offering a 33.6% increase in throughput** on lower FLOPs. This trend holds for larger models, where our VCA-PVT variants outperform stronger Swin Transformer counterparts with **13.0%-23.5% higher throughput**.
>
> ---
>
> 3. **Statistical Significance**
>
>     - To address the concern about statistical significance, we trained the VCA-DeiT-Tiny model three times with different random seeds.
>
>         | Model | FLOPs(G)| Run | Top-1 Acc. |
>         |:---|:---:|:---:|:---:|
>         | VCA-DeiT-T| 1.2 | 1 | 75.5 |
>         | | 1.2 | 2 | 75.5 |
>         | | 1.2 | 3 | 75.4 |
>
>     - The results show minimal variance, with a standard deviation of just 0.05. This demonstrates that the performance gains from VCA are consistent and the training process is stable.
>
> ---
>
> 4. **Visualization of Contrast Maps**
>
>     - Due to updated NeurIPS policies, we cannot include images in this rebuttal. Instead, we provide a textual description of our visualization analysis, as requested.
>
>     -   Visualization Method: We visualize the contrast map in the VCA-DeiT-Tiny model, which has 12 attention blocks. For each block, we pick the head with the largest L2-norm of the Stage-I contrast map. We export the positive $\text{A}^+=\text{Softmax}(\boldsymbol{t}_{+}^{(m)} \boldsymbol{k}^{(m) \top} / \sqrt{d})$ and negative $\text{A}^-$ matrices, compute $\Delta = \text{A}^+ - \lambda \text{A}^-$ , average the $n$ rows, reshape to 2-D grid, upsample to the original image size and overlay it as a heatmap on the original image—highlighting patches that VCA block promotes or suppresses.
>
>     -  **Observations**: In **early blocks** (e.g., block 3), the positive map $A^{+}$ tends to activate on coarse object edges, while the negative map $A^{-}$ activates on background textures. The resulting differential map $\Delta$ often exhibits a classic center-surround pattern, separating the foreground subject from its context. In **late blocks** (e.g., block 10), $A^{+}$ sharpens its focus on semantic parts (e.g., the head and tail of a bird), while $A^{-}$ becomes more diffuse, indicating that distracting correlations have been effectively filtered out.
>
>     -  **Failure Cases**: We observed that the contrastive separation in $\Delta$ is less effective in two main scenarios: (i) **texture-dominated scenes**, such as an image of "leopard skin," where both positive and negative maps activate strongly on the same texture; and (ii) images with **very small objects** that are smaller than the resolution of our pooled grid (e.g., 7×7), making it difficult to generate distinct contrastive views. In these cases, VCA's benefit is diminished.
>
> ---
>
> 5. **Ablation on Pooled Grid Size**
>
>     -  As requested, we analyzed how performance varies with the number of visual contrast tokens ($n$), which is determined by the pooled grid size. This clarifies the trade-off between computational cost and model fidelity.
>
>     -  **Image Classification (VCA-DeiT-Tiny)**: We ablated the number of tokens by pooling the original 14×14 feature map to 5×5, 7×7, and 9×9 grids.
>
>         | Grid $h$×$w$ | $n$ | Params | FLOPs | Top-1 Acc. | Throughput (img/s) |
>         |:---|:---:|:---:|:---:|:---:|:---:|
>         | 5×5 | 25 | 5.9 M | 1.2 G | 75.4 | 4517 |
>         | **7×7 (ours)** | 49 | 6.0 M | 1.2 G | 75.5 | 4494 |
>         | 9×9 | 81 | 6.1 M | 1.3 G | 75.7 | 4263 |
>
>     - The results show a clear trade-off: increasing $n$ yields modest accuracy gains at the cost of slightly more parameters and reduced throughput. We chose **7×7** as our default because it offers a strong balance and corresponds to half the spatial resolution of the input feature map.
>
>     -  **Image Generation (VCA-DiT-S/2)**: We pooled the 16×16 latent map to 4×4, 8×8, and 16×16.
>
>         | Grid $h$×$w$ | $n$ | Params | FLOPs | FID-50k ↓ |
>         |:---|:---:|:---:|:---:|:---:|
>         | 4×4 | 16 | 33.2 M | 5.6 G | 65.5 |
>         | **8×8 (ours)** | 64 | 33.6 M | 5.9 G | 62.1 |
>         | 16×16 | 256 | 33.6 M | 7.3 G | 58.4 |
>
>     - A similar trend appears, where a larger grid improves the FID score. We chose **8×8** for our main experiments, as it again represents half the latent's spatial resolution and provides a compelling trade-off between generation quality and computational cost.
>
> ---
>
> We hope this clarifies our contributions and performance benefits, and we await your feedback.

---

> ### Author Response · Authors · 2025-08-05
>
> Dear reviewer f3ww
>
> Could you kindly spare a moment to review our rebuttal, where we have addressed each of your comments with additional experiments and analyses? Your time and thoughtful feedback are greatly appreciated.
>
> Sincerely, The Authors of Submission 2322

---

> ### Author Response · Authors · 2025-08-07
>
> Dear Reviewer,
>
> I hope this message finds you well.
>
> With the discussion period ending in **less than two days**, we just wanted to gently follow up on our rebuttal. We hope our response has satisfactorily addressed your valuable concerns.
>
> We would be very grateful for your feedback on our clarifications. Please let us know if there is anything else we can address.
>
> Thank you again for your time and invaluable insights.
>
> Sincerely, The Authors of Submission 2322

---

### Official Review · Reviewer_ptg9 · 2025-07-10

**Clarity:** 2
**Significance:** 2
**Originality:** 2
**Rating:** 3
**Confidence:** 1

**Summary:**

As I notified the AC in June 2nd, I deeply apologize but due to an unexpected change in my professional schedule, I will not be able to perform my reviews this year.

**Questions:**

As I notified the AC in June 2nd, I deeply apologize but due to an unexpected change in my professional schedule, I will not be able to perform my reviews this year.

**Ethical Concerns:**

["NO or VERY MINOR ethics concerns only"]

**Final Justification:**

Please ignore my rating. I still don't understand why I have to go through this even though i notified early on that i would not able to perform my reviews this year.

**Limitations:**

As I notified the AC in June 2nd, I deeply apologize but due to an unexpected change in my professional schedule, I will not be able to perform my reviews this year.

**Paper Formatting Concerns:**

As I notified the AC in June 2nd, I deeply apologize but due to an unexpected change in my professional schedule, I will not be able to perform my reviews this year.

**Quality:**

2

**Strengths And Weaknesses:**

As I notified the AC in June 2nd, I deeply apologize but due to an unexpected change in my professional schedule, I will not be able to perform my reviews this year.

---

> ### Author Rebuttal · Authors · 2025-07-31
>
> We sincerely thank the reviewer for their time and completely understand that unexpected professional obligations can prevent a review.
>
> - We noticed that while your comment explains the situation, the submission still includes numerical scores and a final rating of "Borderline Reject." We would like to highlight that this stands in contrast to the detailed and very positive assessments from the other reviewers. They described our work as "an excellent work" and "well-structured," praising its "impressive experimental results," "clear logic," and the "significant performance improvement" achieved without additional computational cost.
>
> - Given this, we would be very grateful if your schedule now permits you to take a look at our paper; the strengths identified by others might be of interest to you as well. If a full review is not feasible, we would kindly and respectfully ask if you might consider adjusting the numerical scores to reflect that a detailed assessment was not performed. Our concern is that the current "Borderline Reject" rating, particularly with a confidence score of 1, could have an unintended influence on the final decision, which would not align with the thorough and positive evaluations provided by the other reviewers.
>
> Thank you again for your time and understanding.

---

### Note · Authors · 2025-08-16

Dear Area Chair and Reviewers,

Thank you for your valuable feedback, which has significantly strengthened our work. We are grateful for the positive engagement and summarize the outcomes below:

* Reviewer f3ww kept their score at borderline accept. We believe we have fully addressed their requests by providing extensive wall-clock/memory benchmarks, multi-seed validation, detailed descriptions for our visualizations, and an ablation on the pooled grid size.

* Reviewer aXpT, initially borderline accept, now supports acceptance, stating, "The newly supplied ablations and clarifications have substantially strengthened the manuscript, resolving my earlier reservations... I am pleased to support acceptance of this work."

* Reviewer Z3tv, initially borderline reject, raised their score, confirming "the rebuttal has addressed most of my concerns." They appreciated our new comparisons and ablation studies.

* Reviewer sZxA, initially borderline reject, modified their score after our rebuttal. We addressed their concerns on practical performance and design by providing detailed wall-clock benchmarks, showing compatibility with kernels like FlashAttention-2, and adding a new ablation to empirically support our pooling mechanism.

Throughout the rebuttal, we substantially enhanced the manuscript:
* **Added hardware performance benchmarks** (latency, memory, throughput) to demonstrate practical efficiency.
* **Conducted new experiments** comparing against four linear attention methods, showing superior performance.
* **Validated on a larger model** (ViT-B) and committed to including ViT-L results in the final version.
* **Provided crucial ablation studies** on pooling methods, hybrid architectures, and other key design choices to solidify our methodology.

With at least three of four reviewers now supporting acceptance (one modified rating is not visible to us), we believe our significantly strengthened manuscript will be of high interest to the NeurIPS community.

Thank you for your consideration.

Sincerely,

The Authors

---

### Decision · Program_Chairs · 2025-09-17

**Decision:**

Accept (poster)

**Comment:**

The submitted paper addresses the attention computation in transformer and replaces the traditional MHA by a new learnable variant of differential transformers, where dense queries are grouped into positive and negative parts. The paper received 4 reviews with ratings which were all above borderline after the author/reviewer discussion phase. One additional 5th reviewer was in the system but couldn't provide a review and the AC was unable to remove them. The corresponding review and rating were completely ignored during the peer review process.

The reviewers appreciated the motivation, clear methodology, the idea of embedding the notion of contrast into an attention mechanism, and experimental validation, with multiple tasks having been tested.

Weaknesses were
- lukewarm gains in concrete speed-ups / exact numbers on this
- Some missing ablations, and analysis of token generation.
- the lack of baselines with linear attention

The authors have provided a rebuttal with new experiments, including detailed wallclock timings, and comparisons with the requested baselines, but also improved the justification of the methodology.

The reviewers agreed that this paper is a valuable contribution to the community and the AC concurs.